

# Fracture dynamics in an unstable, deglaciating headwall, Kitzsteinhorn, Austria

Andreas Ewald[1], Ingo Hartmeyer[2], Markus Keuschnig[2], Andreas Lang[1], Jan-Christoph Otto[1]

[1]Department of Geography and Geology, University of Salzburg, Salzburg, 5020, Austria

[2]Georesearch Forschungsgesellschaft mbH, Wals, 5071, Austria

*Correspondence to*: Jan-Christoph Otto (Jan-Christoph.Otto@sbg.ac.at)

**Abstract.** Processes destabilising recently deglaciated rockwalls, driving cirque headwall retreat, and putting high alpine infrastructure at risk are poorly understood due to a lack of in situ monitoring data. Deglaciation initiates internal stress redistribution and drastically increases atmospheric forcing rendering cirque headwalls particularly prone for rock slope
failure. Here we present quantitative data from an unstable, recently deglaciated cirque headwall. We monitor the dynamics of a fracture at the north face of the Kitzsteinhorn (3203 m a.s.l.) over a period of 2.5 years. Two crackmeters measure horizontal and vertical crack deformation with a resolution of ±0.003 mm and are complemented by crack top temperature measurements. To decipher thermo-mechanical from cryogenic forcing, thermal expansion coefficients for both horizontal and vertical directions are calculated to derive purely thermo-mechanical deformation. Our data shows that fracture
dynamics are dominated by thermo-mechanical expansion and contraction of the inter-cleft rock mass during snow-covered and snow-free periods. Significant deviations from thermo-mechanical behavior occur due to freeze-thaw action during spring and autumn zero curtain periods. Exceptional vertical deformation during these periods is triggered by rainfall events providing liquid water into the fracture system. Subsequent refreezing rather than hydrostatic pressure build-up is to the most likely cause of the mechanical response. Lower magnitude horizontal deformation occurs in autumn and early winter due to
ice segregation. Irreversible fracture opening was not observed, however, enhanced cryogenic deformation in spring and autumn may lead to shallow, low magnitude rock detachments. Our results highlight the importance of liquid water intake in combination with subzero-temperatures on the destabilisation of glacier headwalls. We conclude that intense frost action and ice segregation are common processes in randkluft systems, serving as important preparatory factors of paraglacial rock slope instability.

## 1 Introduction

Glacier retreat is one of the most significant consequences of 20[th] and 21[st] century temperature rise in the European Alps (Haeberli et al., 2007) and has been particularly pronounced in recent decades (Zemp et al., 2015). Retreat is most obvious at glacier tongues but also significant at glacier cirques. Here, glacier retreat causes ice surface lowering and the degradation of ice faces, and thus exposes fresh, frequently oversteepened headwall sections. As headwalls deglaciate, internal stress



redistribution occurs (McColl, 2012) and atmospheric forcing intensifies (Phillips et al., 2017) rendering deglaciating headwalls particularly prone to rock slope failure.

Deglaciation triggered rock-slope instability poses a risk to man and high-alpine infrastructure and is expected to increase in a further warming climate. An increasing number of large-scale rock slope failures following glacier retreat has been reported recently (Huggel et al., 2012, Deline et al., 2009, Allen et al., 2011, Korup et al., 2010, Ravanel et al., 2017). These processes are considered within the concept of paraglacial adjustment and comprise, besides others, oversteepening and debuttressing caused by glacial erosion (Augustinus, 1995). Both cause internal stress redistribution and the propagation of pre-existing discontinuities as well as the development of new fractures (Cossart et al., 2008). The speed of internal stress relaxation depends on rock mass properties (Ballantyne, 2002) as well as on stress history (Graemiger et al., 2017). As stated by McColl (2012), a key challenge in understanding paraglacial rock slope stability is explaining the often-observed lag-times between glacier retreat and rock slope failure. To improve understanding of paraglacial dynamics and related rock slope failure crucially hangs on identifying and quantifying stability-relevant processes in rock slopes.

Instability in shallow bedrock has been detected and quantified by monitoring fracture dynamics in periglacial environments of the Japanese (Matsuoka, 2001) and European Alps (Matsuoka et al., 1997, Matsuoka, 2008, Hasler et al., 2012, Weber et al., 2017, Draebing et al., 2017). Often a negative correlation between temperature and fracture width is observed and attributed to thermal expansion of the inter-cleft rock masses in summer (fracture closing) and contraction in winter (fracture opening) (Wegmann and Gudmundsson, 1999). Thermo-mechanical forcing alone can cause irreversible opening of fractures (Gischig et al., 2011, Bakun-Mazor et al., 2013, Eppes et al., 2016) indicating rock slope instability and may finally result in failure where no obvious trigger is present (Collins and Stock, 2016). This process is often superimposed and reinforced by cryogenic pressure build-up and release in the rock matrix, e. g. by ice segregation (Weber et al., 2017), and can also initiate new cracks (Matsuoka and Murton, 2008).

At various sites, the thermo-mechanical regime alone cannot explain fracture dynamics (e. g. Hasler et al., 2012). In the Swiss and Japanese Alps, two peaks of exceptional crack widening in spring and autumn are reported (Matsuoka, 2001, 2008 and Matsuoka et al., 1997). Highest deformations occur episodically during spring zero curtain periods and are explained by downward migration and refreezing of meltwater confined between snow cover and frozen bedrock (Matsuoka et al., 1997, Matsuoka, 2001). In a long-term observation, Matsuoka (2008) shows that the permanent crack enlargement over a period of ten years was mainly caused by two spring events. As soon as daily rock temperature oscillations indicate snow-free conditions, fracture dynamics return to thermo-mechanical forcing. Net expansion can be further caused by autumn freezing of in situ water (Matsuoka, 2001), ice segregation (Draebing et al., 2017) or hydro-thermally induced strength reduction of ice-filled joints (Hasler et al., 2012, Weber et al., 2017). Ice relaxation and ice extrusion can compensate cryostatic pressure build-up (Davidson and Nye, 1985), even though cyclic repetition of thermo-cryogenic deformation leads to fatigue and ultimately rock slope failure (Jia et al., 2015).



Previous findings commonly highlight the importance of moisture availability as well as rock temperature for timing and magnitude of frost shattering (Matsuoka et al., 1997) and support the efficiency of frost wedging in glacier headwalls (Matsuoka, 2001, Gardner, 1987, Sanders et al., 2012).

Here, we investigate the deformation regime of an open fracture, which is of direct geotechnical relevance for a popular cable car station. The fracture is situated immediately up slope of the detachment zone of a recent rockslide (2012) (Keuschnig et al., 2015) and was glacially covered until the 1980s. The present study focuses on the need for quantitative data from unstable, recently deglaciated rock slopes, essential for better understanding the increasing risk on high-alpine infrastructure. Based on a 2.5-year monitoring campaign, this study aims to decipher and quantify stability-relevant processes and their temporal occurrence, and addresses the following research questions:

- Are fracture dynamics dominated by thermo-mechanical expansion/contraction of the inter-cleft rock mass?
- Do cryogenic processes, i.e. freeze-thaw dynamics and ice segregation, affect fracture opening/closing?
- Can irreversible crack deformation patterns and destabilisation be observed?

We show how liquid water intake in combination with subzero-temperatures leads to significant fracture deformation. During spring and autumn zero curtain periods a freeze-thaw window between -1 °C and 0 °C exists in which the general thermo-mechanical deformation regime is overprinted. Evidence for active ice segregation is found in a temperature window between -4 °C and -8 °C. Rock anisotropy due to foliation favours frequent, small, shallow rock detachments whereas fracture widening leading to larger magnitude rock slides is rare.

## 2 Study area

The Kitzsteinhorn (3203 m a.s.l.), Hohe Tauern Range, in the State of Salzburg, Austria (Fig. 1a) was chosen as study site. Its dominant summit pyramid (Fig. 1b) is part of the central Alpine chain and prominently towers over the surrounding peaks and ridges. The studied north-facing headwall stands about 100 vertical meters tall above the current glacier margin (~2940 m a.s.l.) up to the Kitzsteinhorn west ridge (Fig. 1c). The mean inclination of the slope is around 45°. In its lowest section the slope is vertical and forms the glacier randkluft system.

The rock wall has developed in calcareous mica-schists bearing quartz veins. The rock is heavily jointed and foliated. Major joint sets K1 and K2 (Fig. 1d) dip perpendicular to the slope in W and SW direction, respectively. Minor joint sets dip slightly to SSE and deeply to NW (Hartmeyer et al., 2012). The slope has developed parallel to the foliation dip 45° N. Laterally limited by joint set K1 or local escarpments within the rock slope, the joint sets divide the rock mass into cubic to rhomboidal blocks of more than 5 m length. The majority of these blocks is assumed to be held in place solely through cohesion and friction at the shear planes or through mechanical abutments.

The monitored fracture (Fig. 1e) is part of joint set K2 situated in the middle of the slope. It has an aperture of about 5 cm and a large lateral persistence of at least 10 m (Fig. 1f). Fractures of joint set K2 are filled with coarse rock fragments in a fine-grained matrix.

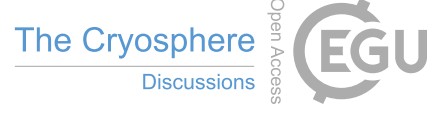

**Fig. 1:** The Kitzsteinhorn is situated in the Hohe Tauern Range, Austria (a). The slopes of the summit pyramid (b) are recently undergoing rapid deglaciation. On the north face, the glacier released a cataclinal rock wall (c) with joint sets K1 and K2 perpendicular and foliation (SF) parallel to the surface (d). Deformations of a fracture of joint set K2 are monitored in horizontal ($CD_H$) and vertical ($CD_V$) direction (e). The headwall material mainly consists of highly weathered, discontinuous calcareous mica-schist (f).

## 3 Methods

### 3.1 Data Acquisition

For permanent monitoring of fracture dynamics, two Geokon Model 4420 Vibrating Wire Crackmeters were installed to measure horizontal ($CD_H$) and vertical crack deformation ($CD_V$) (see Fig. 1e). $CD_H$ detects the opening and closing pattern of the fracture, $CD_V$ the distance between the two inter-cleft rock masses relative to each other. Grouted anchors fix the ends of the crackmeters and a metal casing and a metal plate protect the instruments from damage through snow and rockfalls. Crackmeter resolution is ±0.003 mm for $CD_H$ and ±0.01 mm for $CD_V$. Air temperature as proxy for near surface rock temperature is recorded using integrated thermistors directly above the fracture (crack top temperature, CTT).

Air temperature (AT) and snow height data is acquired from the high-alpine meteorological station "Glacier Plateau" (2940 m a.s.l.), located less than 300 m from the monitored fracture in the center of the Schmiedingerkees glacier. Precipitation is recorded at the meteorological station "Alpincenter" (2450 m a.s.l.). Pt1000 thermistors with an accuracy of ±0.03°C, installed in a 30 m deep borehole close to the observed fracture, provide rock temperatures (RT) at depths of 2, 3 and 5 meters. Temperature readings are taken every 10 minutes.

### 3.2 Data Analysis

Crackmeter values are converted into distance values and corrected for temperature variations. For data reduction, hourly mean values for crackmeter and rock temperature are calculated. The presence of snow cover is derived from CTT data, following the approach of Haberkorn et al. (2015): snow cover is present when the standard deviation of CTT is < 0.5 °C. Due to significant differences in inclination and wind exposure the snow height record from the weather station "Glacier Plateau" is not representative for the study site.

Expansion and contraction of the inter-cleft rock mass is derived using the approach of Bakun-Mazor et al. (2013), as recently applied in a periglacial environment by Draebing et al. (2017). Due to strong rock anisotropy, coefficients for both, horizontal ($\alpha_H$) and vertical ($\alpha_V$) deformation components are calculated. The expansion coefficients $\alpha$ for horizontal and vertical deformation components can be expressed by:

$$\alpha_H = \frac{-\Delta CD_H}{2 \mathrm{L}\, \Delta CTT} \quad (1)$$

$$\alpha_V = \frac{-\Delta CD_V}{\mathrm{L}\, \Delta CTT} \quad (2)$$

where ΔCD is the 10-day mean crack deformation, ΔCTT is the 10-day mean crack top temperature and L is the mean block length (Fig. 2). As an approximation, $CD_V$ is taken to represent the deformation of one single block. In the case of $CD_H$, the





deformation relates to two blocks and consequently L has to be multiplied by two. To convert rock deformation into fracture deformation the results are multiplied by minus 1.

To unravel thermo-mechanical from cryogenic fracture deformation, α is modified by considering only periods where CTT remains below -10 °C. For distinction the modified coefficient is referred to as α'. We assume that fracture deformation at CTT below -10 °C is governed exclusively by thermo-mechanical controls because (i) fracture deformation by ice segregation is of minor importance, (ii) freeze-thaw dynamics can be ruled out, and (iii) cleft infill is unlikely to inhibit fracture closing. Thermo-mechanically induced fracture deformation is thus modelled by:

$$\Delta CD_H = -\alpha'_H 2\mathrm{L}\ \Delta CTT \quad (3)$$

$$\Delta CD_V = -\alpha'_V \mathrm{L}\ \Delta CTT \quad (4)$$

The modified coefficient α' is later used to compare measured CD values to CD values expected if movement is exclusively due to thermal expansion/contraction allowing to differentiate between different crack deformation controls.

**Fig. 2:** Dimensions of the inter-cleft rock mass are acquired from geotechnical mapping and optical borehole scans. Fracture dynamics are assumed to be primarily driven by thermal expansion and contraction of adjacent rock blocks as well as by cryostatic pressure.

## 4 Results

### 4.1 Meteorological and Rock Temperature Data

Mean annual air temperature (MAAT) at the local weather station ("Glacier Plateau") was -1.7 °C in 2016 and -2.7 °C in 2017. AT shows daily amplitudes of up to 10 °C throughout the season (Fig. 3a). Mean annual rock temperatures (MART) in depths of 200 cm, 300 cm and 500 cm all stayed below 0 °C (Fig. 3b). Despite lower MAAT in 2017, maximum active layer depth increased from 3.0 m in 2016 to 3.6 m in 2017 due to high summer temperatures. Two thermal anomalies were observed during the spring zero curtain periods in 2016 and 2018, as RT300 abruptly increased by about 1 °C within one hour (highlighted using arrow symbols in Fig. 3b). Along with a shorter zero curtain period, this anomaly was absent in spring 2017 and was also not detected at other depths. During the observation period, zero curtain periods varied from May to June, with no obvious correlation to snow cover thickness. In contrast to AT, CTT only showed large diurnal fluctuations above 0 °C (Fig. 4). In the subzero range, daily CTT fluctuations did not exceed 1 °C. Mean annual crack top temperature (MACTT) was more than 1 °C lower (-3.3 °C) than MAAT in 2016. In 2017, MACTT decreased to -3.7 °C. Maximum CTT (13 °C) was recorded in August 2017, minimum CTT (-15 °C) in January 2017, spanning an overall range of 28 °C.

**Fig. 3:** Meteorological (a) and rock temperature data (b) from January 2016 to July 2018. Light grey background indicates snow cover (Haberkorn et al., 2015), dark grey indicates zero curtain periods. Arrows highlight abrupt increases in rock temperature at 3 m depth, indicating water percolation in the joint system.



## 4.2 Crack Deformation Data

Amplitude of annual fracture deformation was 1.4 mm for horizontal measurements ($CD_H$), and 5 mm for vertical measurements ($CD_V$). During summer in 2016 and 2017 vertical deformation exceeded the contraction limit of the crackmeter so only minimum values for $CD_V$ were established. For most of the time, a negative correlation between CD and

CTT is observed in both, the horizontal and the vertical deformation component. Deviations from the negative correlation are mostly associated with sustained zero curtain periods. Fracture opening shows a pronounced seasonal cycle (Fig. 4a) with a maximum during minimum CTT and a minimum during summer, around August.

During periods of snow cover (highlighted in grey in Fig. 4), crack deformation shows insignificant daily fluctuations, whereas during snow-free conditions (white background, Fig. 4) daily deformation changes up to 0.2 mm occur. $CD_H$

generally correlates well with CTT under snow-covered conditions. Deviations from this general trend occurred in early October 2016 during the autumn zero curtain period. Here, instead of opening the crack closed by 0.2 mm as AT approached -10 °C. A similar reverse pattern is also observed in autumn 2017, when the crack closed by about 0.1 mm. Generally, trends in vertical deformation follow trends in $CD_H$ during snow-covered periods (Fig. 4b).

During the zero curtain period in June 2016 the $CD_V$ trajectory suddenly changed from a decreasing trend to a rapid increase

of 0.4 mm within two weeks (see label "I" in Fig. 5a), followed by a sudden decrease of at least 0.9 mm within 24 hours (see label "II", Fig. 5a) when the snow cover disappeared. The the remarkable knick point in the $CD_V$ trajectory on June 8th 2016 matches with the knick point in the RT300 curve, which was probably triggered by water infiltration after a significant rainfall event (6 mm h$^{-1}$, June 8th 2016, 04:00 PM). During other rainfall events of similar magnitudes within the zero curtain period however, no similar increases in $CD_V$ can be observed in spring 2017 (Fig. 5b). An abrupt increase of CTT in the

absence of rainfall occurred prior to the zero curtain period (see label "I", Fig. 5b) in spring 2017 as well as a sudden decrease of $CD_V$ at the end of the zero curtain (see label "II", Fig. 5b). In spring 2018, an abrupt increase of CTT and $CD_V$ is observed on May 12th without preceding precipitation (see label "I", Fig. 5c). The subsequent $CD_V$ increase progressed relatively slow and was followed by a stepwise decrease (see label "II", Fig. 5c). A long-lasting, high-magnitude rainfall event later triggered a 2 mm increase (see label "III", Fig. 5c), which again was followed by a stepwise decrease of more

than 3 mm at the end of the zero curtain period (see label "IV", Fig. 5c).

**Fig. 4: O**bserved crack top temperature with (a) horizontal and (b) vertical crack deformations. Data gaps mainly appeared as the vertical crackmeter device went beyond the lower measurement limit during summer 2016 and 2017 until it broke down and had to be replaced (*) in October 2017. Values then had to be reset to zero.


A sudden increase of the 3 m rock temperature on May 26th does not match with precipitation input and no mechanical change was detected at that time by the crackmeter devices. A scatter plot of horizontal versus vertical deformation (Fig. 6) reveals two movement patterns: During snow-covered periods kinematics are dominated by deformation parallel to the





terrain surface (see label "I", Fig. 6) whereas zero curtain periods result in significant vertical deformation (see label "II", Fig. 6).

**Fig. 5:** Deformation regime, crack top- and rock temperature at 3 m depth as well as hourly precipitation during spring zero curtain periods 2016 (a), 2017 (b) and 2018 (c). Sudden increase of CDV is obviously triggered by precipitation events (1) and decreases abruptly when zero curtain periods end (2).

**Fig. 6:** Two-dimensional fracture deformation showing the position of the lower block relative to the upper. The separation of the two graphs is due to the new crackmeter installation in September 2017. Colours indicate snow-covered periods, consistent with crack top temperatures below -1 °C (blue), zero curtain periods between -1 and 0 °C (green) and snow-free conditions above 0 °C (red). Two distinct movement patterns (I) parallel to the surface during snow-covered periods and (II) in vertical direction during zero curtain and snow-free periods are identified.

## 4.3 Crack Deformation Modelling

To model thermo-mechanical crack deformation, thermal expansion coefficients (α') are calculated. During periods with CTT below -10 °C the thermal expansion coefficient α' is $7 \times 10^{-6}$ °C$^{-1}$ horizontally and $14 \times 10^{-6}$ °C$^{-1}$ vertically demonstrating pronounced anisotropic behaviour. Near surface thermal expansion of calcareous mica-schist perpendicular to foliation is twice the expansion parallel to foliation. Modelling of thermo-mechanical crack deformations based on α' is in good agreement with measured deformations (Fig. 7). In winter, differences between measured and modelled horizontal deformation occur immediately after maximum opening (see arrow "I", Fig. 7a). During late winter and spring, the gap between measured and modelled deformation remains constant at about 0.3 mm. Modelled summer deformation shows five times higher daily fluctuations (1 mm) than measured deformation (0.2 mm) (see arrow "II", Fig. 7a). In autumn (2016 and 2017) the curves re-join after temporarily deviating from the negative temperature dependency during the zero curtain period (see arrow "III", Fig. 7a). The modelled vertical deformation is highly consistent with the measured deformation during snow-covered periods but significant deviations occur during spring zero curtain periods (see arrow "IV", Fig. 7b). Modelled summer expansion shows daily fluctuations of approximately 0.3 mm. In autumn 2016 modelled deformation does not follow the measured CD$_V$ increase that is much faster than thermo-mechanically predicted (arrow "V", Fig. 7b).

**Fig. 7:** Measured and modelled (a) horizontal and (b) vertical deformations. Deviations from purely thermo-mechanical regime occur right after peak opening in winter (I). Daily summer fluctuations are largely overestimated (II) but graphs re-join during autumn zero curtain (III). Vertical deformations are consistent with thermo-mechanical forcing during snow covered periods whereas significant deviations occur during spring (IV) and autumn zero curtain periods (V).





## 5. Discussion

### 5.1 Data Quality and sources of uncertainty

Owing to the extreme environmental conditions in which the monitoring was carried out a number of assumptions need to be considered when interpreting the results. First, for interpretation homogeneous rock mass conditions within a block are assumed and micro-fissures and other inhomogeneities that may affect deformation regimes at the monitored fracture are ignored. Second, crack top temperatures may not represent the entire fracture. A protective metal casing shielded the devices from mechanical forces e. g. rockfall and snow load. Observed temperature may thus vary from actual environmental conditions as thermistors are integrated in the crackmeter devices housing and covered below the metal casing,. Third, crackmeters provide point data only and thus the vector of motion remains uncertain. With a point data set, it is not possible to determine the mode of dislocation and differentiate between toppling, sliding, or rotational movement. Furthermore, as the vertical crackmeter had to be replaced in October 2017, subsequent $CD_V$ values cannot easily relate to the preceding data.

### 5.2 Identifying Deviations from Thermo-mechanical Deformation

In the present study a significant negative correlation between crack opening and temperature is detected and reflects a pronounced thermo-mechanical control on crack deformation. This deformation regime is similar to patterns observed in previous studies of high-alpine, periglacial environments (Matsuoka, 2001, Hasler et al., 2012, Weber et al., 2017, Draebing et al., 2017). Thermal expansion coefficients calculated for horizontal and vertical deformation are also within the range of values presented in previous studies (Cooper and Simmons, 1977, Robertson, 1988, Hasler et al., 2012).

To better demonstrate the temperature-dependence of CD, expansion coefficients ($\alpha_H$, $\alpha_V$) are plotted against the full range of observed (averaged) CTT (Fig. 8). This analysis reveals that $\alpha$ mainly clusters within a relatively narrow corridor that does not show significant variations with temperature. This corridor likely reflects thermo-mechanical deformation and is therefore referred as 'thermal expansion window' (Draebing et al., 2017). For $CD_H$, coefficients ($\alpha_H$) from 0 to 25 x $10^{-6}$ °$C^{-1}$ (Fig. 8a), and for $CD_V$, coefficients ($\alpha_V$) from 0 to 50 x $10^{-6}$ °$C^{-1}$ (Fig. 8b) are considered as reflecting the 'thermal expansion window'. Cases that significantly deviate from thermo-mechanical behaviour are discussed in the following.

**Fig. 8:** 10-day mean expansion coefficient $\alpha$ plotted against 10-day mean crack top temperature reveals temperature-dependent horizontal (a) and vertical crack deformation (b) within a thermal expansion window (TE; yellow), freeze-thaw window (FT; light blue) and ice-segregation window (IS; blue) (after Draebing et al., 2017).

### 5.2.1 Freeze-Thaw Action

Significant deviations of $\alpha$ from values within the 'thermal expansion window' occurred when temperatures fell just below freezing approximate between 0 and -1 °C. Patterns observed in this thermal window are likely to be caused by freeze/thaw action and the associated volume change of water ('freeze-thaw window', Draebing et al., 2017).





Freeze-thaw-induced variations of α were significantly more pronounced for $CD_V$ (Fig. 8b), than for $CD_H$ (Fig. 8a), and occurred mainly during distinct zero curtain periods. In spring and autumn of 2016 and 2018 significant differences between measured and modelled $CD_V$ are observed: during zero curtain periods increase in measured values outpaced thermo-mechanical predictions (Fig. 7b).

During spring zero curtain periods the subsurface is still frozen, while rainfall and snow-melt provide substantial amounts of liquid water to the rock mass. The abrupt rise in rock temperature at 3 m depth in 2016 and 2018 is probably caused by fluid flow in the joint system and thus indicates liquid water availability. Partial near-surface (re-)freezing of available rain-/melt-water within surface-parallel foliation interstices may therefore be responsible for the significant $CD_V$ increase that can be observed during these periods. As soon as the snow cover fully disappears, ice in the near surface rock mass melts, probably

causing the observed rapid contraction at the end of the zero curtain period (Fig. 7b). In spring 2017, no enhanced vertical expansion was observed before rapid, melting-induced contraction. Even though some rainfall was recorded moisture availability may still have been limited and snowmelt may have been too rapid for near-surface melt-water refreeze, as observed by Matsuoka et al., (1997). Missing initiation of new cracks during winter that later provide space for refreezing water may also play a role.

The temperature changes observed during spring zero curtains at 3 m depth represent a significant deviation from otherwise uniform ground thermal conditions. In spring 2016, the abrupt rise in rock temperature at 3 m depth matched with rapid $CD_V$ changes and was most likely caused by an intense rainfall event. In spring 2018 no significant mechanical reaction was observed during temperature rise in 3 m depth, which was probably induced by meltwater flow as rainfall was absent, similar as described by Hasler et al., (2011). Differences in the intensity of water intake or variations of the physical state of the

fracture system could explain the observed inter-annual $CD_V$ differences between 2016 and 2018. It is also possible that fluid flow in the fracture system is of subordinate importance for mechanical responses, as physical processes that act close to the rock surface may dominate $CD_V$.

Comparable to spring time, significant $CD_V$ increases were also observed during autumn zero curtain periods. A substantial $CD_V$ rise was recorded in October 2016 (Fig. 4b, 7b), probably related to freezing of rainwater within the surface of the

cooling rock mass. Fissures and microcracks in the highly weathered bedrock are likely to provide sufficient space for water storage that can cause the observed mm-range expansion during autumn freezing.

For $CD_H$, where freeze-thaw deviations were significantly less pronounced than for $CD_V$ (Fig. 8a), clear deviations between measured and modelled values were mainly found during the 2016 autumn zero curtain period, and to a lesser degree also during the 2017 autumn zero curtain period. During these periods the thermo-mechanical $CD_H$ increase induced by autumn

cooling is interrupted by a significant closure of the fracture (Fig. 4, Fig. 7). The causes behind this pattern remain elusive but may be related to active layer freezing (Fig. 3b) that mainly happens top-down from the surface and only to a lesser extent bottom-up from the permafrost table (Dobinski et al., 2011). Commonly a considerable hydro- and cryostatic pressure is built up within the remaining active layer (Jia et al., 2017). While ice extrusion can compensate for these pressures (Davidson and Nye, 1985), they may still be sufficient to cause fracture deformation especially if no intact rock bridges



remain. In this case, volumetric expansion may efficiently lead to fracture deformation, overcoming the strength of mechanical abutments or frictional strength of rock-rock contacts on the shear plane (Krautblatter et al., 2013). If one fracture is considered in isolation, increasing cryo-/hydrostatic pressures in the freezing cleft infill would result in horizontal fracture opening. The observed closing of $CD_H$ may result from horizontal fracture deformation in neighbouring parts of the

slope system. Local, fracture-scale effects could therefore be overprinted or even reversed by reactions of the slope system, particularly during dynamic periods such as active layer freezing in autumn.

### 5.2.2 Ice Segregation

While not as pronounced as in the 'freeze-thaw window', deviations from the thermo-mechanical deformation regime were also found at lower CTT, between -4 °C to -8 °C, which is consistent with an 'ice-segregation window' (Hallet et al., 1991,

Anderson, 1998, Murton et al., 2006). At which temperature range ice segregation efficiently occurs is subject to recent debate (Matsuoka and Murton, 2008), lab experiments (Duca et al., 2014), and geophysical field experiments (Girard et al., 2013). Based on experimental studies, Hallet et al. (1991) suggest a temperature window between -3 °C and -6 °C, whereas based on field monitoring Draebing et al. (2017) found evidence for ice-segregation related deformation on a much wider range between -1,4 °C and -9,3 °C. Our data points to active ice-segregation during autumn and early winter when CTT is

between -4 °C to -8 °C. Magnitude of deformation, however, is low compared to volumetric expansion within the freeze-thaw window. The inter-cleft rock mass can host supercooled liquid water whole year round (Hall, 2004). Ice-segregation may thus occur at any time if certain sub-zero temperatures prevail. Snow cover plays an important role in preserving these conditions and promoting ice-segregation induced fracture opening (Draebing et al., 2017).

Comparing measured and modelled horizontal deformation in late January and February 2016 (Fig. 7a) shows that horizontal

fracture closing is attenuated after maximum opening. Maximum opening is attributed to thermal contraction of the rock mass and creates space for water migration. Subsequent horizontal fracture closing due to thermal expansion is then potentially hampered by newly formed cleft ice. This process coupling is also assumed to occur during autumn freezing. Thermally induced fracture opening can thus create space for segregation ice growth which, in turn, may promote further fracture opening or at least impede closing. In other words, compound thermo-cryogenic processes may create positive

feedbacks reinforcing fracture growth on a seasonal scale. The efficiency of ice segregation at depth, which is assumed to play a vital role above the permafrost table can, however, not be detected with the current measurement setup.

### 5.2.3 Other Mechanisms

Deviations from thermo-mechanical deformation were also observed outside the 'freeze-thaw window' and the 'ice segregation window'. During warm, snow-free periods in summer, modelled $CD_H$ temporarily exceeded measured $CD_H$ by a

factor of five (Fig. 7a). Overestimation of conductive heat transport efficiency within the rock mass may provide a plausible explanation for the observed deviation. Modelling is based on winter conditions and thus assumes little thermal variation within the investigated block. This is unlikely given that diurnal warming and cooling only penetrates to a depth of a few



decimetres. Modelled deformation may thus lead to an overestimation particularly on days with pronounced diurnal CTT cycles.

Fracture fill potentially inhibits free expansion of the rock mass, i. e. fracture closing, and may thus contribute to deviations between measured and modelled $CD_H$. While fine-grained fill material may deform easily in an unfrozen state, larger fragments wedged in the crack can efficiently prevent expansion. Ultimately, this mechanism can lead to irreversible opening of fractures (Bakun-Mazor et al., 2013) which has, however, not been observed in this study. Occasionally, secondary carbonate precipitates are found within the monitored fracture. Such carbonates bond and stabilise the adjacent rock masses (Viles, 2013) and may provide another reason to subdue fracture dynamics.

## 5.3 Implications for rock slope stability

The headwall system considered in this study comprises approximately 100 vertical meters of frozen rock, extending from the ridge down to the randkluft. The general inclination of the slope at about 45° matches the dip of foliation of the calcareous mica-schist. Instability is generally preconditioned by glacial erosion undercutting the slope and leading to an oversteepening of the lower part of the rock wall (Augustinus, 1995). Recent instability is presumably initiated by deglaciation of this part of the headwall where foliation dips out of the slope and favours sliding failures. Resulting debuttressing is considered a preparatory factor of a recent rockslide of considerable magnitude (Keuschnig et al., 2015). Occurrence of rock slope failures therefore is the expression of paraglacial adjustment (Ballantyne, 2002). In addition, accompanying changes in ground thermal conditions such as permafrost active layer deepening or increased liquid water availability amplify instability (Deline et al., 2015) by decreasing intact rock strength (Mellor, 1973) and increasing temperature-dependent fracture dynamics (Krautblatter et al., 2013).

Thermal expansion and contraction alone can lead to thermal fatigue (Hall and Thorn, 2014) and thus prepare and trigger rock slope failures (Gischig et al., 2011, Collins and Stock, 2016, Eppes et al., 2016, Eppes and Keanini, 2017). Additional cryogenic forcing enhances deformation on weekly to seasonal timescales. Volumetric expansion has been identified as the major process enhancing vertical deformation in spring and autumn and is thus assumed to be the primary driver of shallow, low magnitude rock detachments.

The annual magnitude of horizontal fracture deformation amounts to 1.2 mm, daily deformations in summer do not exceed 0.2 mm. The contribution of an individual fracture to the stability of the entire headwall may thus seem negligible. Assuming similar behaviour, of the neighbouring K2 fractures and extrapolating the observed dynamics across the slope results in annual deformations of more than a centimetre. Corresponding pressures will accumulate downslope, tackling the lowest blocks right above the glacier. The opening of one fracture may, however, also lead to the closing of another, due to large spatial variability of environmental forcing or the kinematic dominance of a single fracture. This could, for instance, explain the fracture closing observed during autumn zero curtain 2016 and 2017, when freezing-induced opening was expected.

Our data shows that temperature in combination with moisture availability is the critical factor determining enhanced fracture deformation on diurnal to seasonal scales (Matsuoka, 2008). The temperature range where most significant





deformations occur lies within a freeze thaw window between -1 °C and 0 °C. The critical timing is when liquid water meets frozen bedrock mainly during zero curtain periods. During spring zero curtain, snow cover protects the rock surface from solar radiation while meltwater percolates and refreezes in the surficial bedrock layers.

Protected from solar radiation and daily/seasonal temperature extremes a similar setting may exist in the randkluft, which represents the lowermost part of the investigated headwall system. Surface run-off can enter the randkluft system during the whole summer period and refreeze due to sustained negative rock temperatures inside the randkluft (Gardner, 1987, Sanders et al., 2012, Hartmeyer et al., in preparation). Conditions in randkluft systems may thus favour intense frost action (Matsuoka 2001) and ice segregation (Sanders et al., 2012), and thus serve as important preparatory factors for subsequent slope destabilisation.

As glaciers are wasting down due to recent atmospheric warming, the lowest headwall sections are exposed to direct atmospheric forcing. Their formerly constant thermal regime is now disturbed by diurnal to annual temperature cycles, a transition recently termed 'paraglacial thermal shock' (Graemiger et al., 2018). A related long-term terrestrial LiDAR study from the Kitzsteinhorn found dramatically increased rockfall activity in recently deglaciated headwall sections and thus confirmed a pronounced paraglacial response (Hartmeyer et al., in preparation). While further measurements are certainly required to elucidate paraglacial crack deformation patterns, the data presented here support the hypothesis that glacier headwalls are particularly prone to new, shallow rock damage during transition from glacial to non-glacial conditions.

## 6. Conclusions

The aim of this study was to decipher and quantify stability-relevant processes and their temporal dynamics in deglaciating headwall systems. Based on 2.5 year monitoring of crack deformation in a north-facing glacier headwall, our data reveals that:

- fracture dynamics are dominated by thermo-mechanical expansion and contraction of the inter-cleft rock mass during snow-covered and snow-free periods. Thermal expansion coefficients of $7 \times 10^{-6}$ °C$^{-1}$ along and $14 \times 10^{-6}$ °C$^{-1}$ perpendicular to foliation highlight strong anisotropy of the calcareous mica-schist.

- Significant deviations from the thermo-mechanical deformation regime occur mainly during spring and autumn zero curtain periods due to freeze-thaw action. Lower magnitude deviations arise in autumn and early winter probably due to segregation ice growth. Besides cryogenic processes, other mechanisms may affect fracture dynamics such as wedged rock fragments impeding maximum fracture-closing during snow-free periods.

- Irreversible fracture opening as precursor of high magnitude rock slope instability was not observed. Instead, enhanced cryogenic deformation in spring and autumn may lead to shallow, lower magnitude rock detachments.

Our results highlight the importance of liquid water intake in combination with subzero-temperatures on the destabilisation of glacier headwalls. Randkluft systems may favour intense frost action and ice segregation, serving as important preparatory factors of paraglacial rock slope instability.





*Competing interests.* The authors declare that they have no conflict of interest.

*Author contribution..* AE processed and analysed the data set. IH carried out the field installations. All authors discussed the results and contributed to the writing and editing of the manuscript.

*Acknowledgements*. This study was co-funded by the 'Austrian Academy of Sciences' (ÖAW) (Project 'GlacierRocks'). We furthermore thank the 'Gletscherbahnen Kaprun AG' for logistical assistance and the 'Geodata ZT GmbH' for technical support.

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
