# Peer review of "Fracture dynamics in an unstable, deglaciating headwall, Kitzsteinhorn, Austria"

_The Cryosphere, 2019_

## Referee Comment (RC1) · Robert Kenner (Referee) · 20 May 2019

**Overall impression**

The paper presents a valuable dataset and should be considered for publication. The manuscript needs some medium reworking to become suitable for publication. In my detailed comments (especially the longer ones) there are some issues with regard to content, which I would like the authors to spend some more thoughts on. Furthermore, I have the feeling that the significance of the paper could be clearly increased if the authors establish a link to the really central question: What is role of deglaciation? I miss this painfully in the paper. Currently the paper presents an interesting dataset from a deglaciered headwall. That's nice and as there are not excessively many comparable datasets it is sufficient for a publication in TC. However, to stick out and become a really important paper the authors have to interpret their results with respect to glaciation. Are the observed processes particularly related to a deglaciered rock wall or can I observe the same in the upper parts of the rock wall which was never covered by glacier ice? What is difference to rock wall kinematics below the glacier line or in the Bergschrund? Is the erosion faster above or below the Bergschrund? In particular the last question is considered by too many authors as obvious but the answer is not obvious at all (see last and third last detailed comment).

I think the paper should at least try to discuss these questions to bring them into the scientific focus. Hereby, I would like to encourage the authors to be more critical against apparently established knowledge what sometimes actually misses sound scientific basis.

I am looking forward to read the new version of the manuscript and hope for a successful publication in TC.

All the best,

Robert Kenner

**Detailed comments:**

**Abstract:**

I miss some information about permafrost presence or absence. This is an important factor that should be mentioned in the Abstract. Searching "permafrost" in the rest of the document gave no clear answer on this question neither.

**Introduction:**

**P1 l29:** This is an interesting (and correct) description but do you have some explanations or references why these headwalls are often oversteepened? I think there is a term for it called schrundlines. See e.g. sanders et al 2012 (cited by you). These schrundlines have obviously developed during glaciation and not after glaciation or during deglaciation. Any thoughts on that?

**P2 l6ff:** I would distinguish these processes more carefully. Debuttressing is probably not a driving factor at headwalls but is more related to lateral (valley) slopes of tongue shaped glaciers. Oversteepening

occurs at headwalls as well as on lateral slopes but the type of glacial erosion is very different between both locations. (Perhaps rather a type of plucking behind the bergschrund!? See again sanders et al.) If you want to focus on headwalls, I think it is important to go more into detail here.

Furthermore, I feel it is a pity that all rock slope failures taking place in the vicinity of a glacier are lumped together by most of the studies in this field. Oversteepening is obviously a result of glacier erosion and not of glacier retreat. Rock slope instabilities caused by oversteepening are thus just secondarily or not related to glacier retreat. This is probably different for rock slope failures whose kinematic was not related to oversteepening or which were previously covered by glacier ice in large parts (especially the discontinuities) as it is the case at your study site.

Just recently we observed a rock fall at Flüela Wisshorn in Switzerland where more than 250'000 m3 collapsed in an old glacier cirque (you can google some pictures of it). This cubature was never covered by glacier ice, not even during the last cold period. However, the cubature was kinematically free, as the release plane (dip slope) cropped out below the cubature already before the event. This was because the lower end of the cubature was built by a terrain step which was part of a distinct schrundline.  In such a case it makes absolutely no sense for me to talk about a slope failure related to glacier retreat. This instability originates most likely from glacial erosion during the last glacial maximum and collapsed now, several decades after the LIA glaciation and several thousands of years after its initiation.  I claim it would have collapsed as well if the LIA glacier below the cubature would still have been present.

Maybe you can consider those differences somewhere in the introduction and perhaps also in the discussion

**P3 l13:** This sounds more than abstract or conclusions. I would not let the cat out of the bag here

**Study area**

**P3 l21:** the….headwall reaches a height of 100 m above the current…

**Data Acquisition**

**P4 l16**: Why just in 3 depth if the borehole is 30 m deep? Why not deeper and why not closer to the surface (to track freezing fronts)?

**Data Analysis**

**P5l5: "**(i) fracture deformation by ice segregation is of minor importance"  Can you give an explanation for this? CTT are a weak indicator for the temperature profiles in greater depth aren't they?

**4 Results**

**P5 l22** Mention presence of permafrost

**P5 l21** active layer depth = linear interpolation between thermistors??

**Figure 3 and text**: What is the zero curtain period you are talking about? How is it defined? I do not get it.

**Figure 3:** The rock temperatures show distinct zero curtains in Autumn but not in spring. This is interesting and important. Is this somewhere discussed? There must be a lot of water somewhere in the rock that caused the long autumn zero curtain by freezing...? But why is there no ZC in spring? Where has the ice gone!? Or has the water percolated away in autumn without freezing??

**P5 l22** Anomalies during zero curtain period? You are talking about ZC at which depth? Increase of 1K during zero curtain at 3m depth means +1°C what is not possible as active layer depth was 3m in 2016!?

Along with shorter zero curtain period… Where? When? Unclear!

Section is hard to follow, try to formulate more precise.

**P6 l16** The the

**P6 l16** Probably because of ice layers at the base of the snow cover which often prevent water infiltration during snow melt. See Phillips *et al.* (2016)

**P6 l33** deformation parallel to the surface means CDh, i.e. horizontal deformation, right? Better use a consistent terminology.

**4.2 Crack Deformation Data** here and in the entire paper: it is confusing if you are talking about CD/CV increase or decrease. Increase or decrease means the cracks deforms faster or slower but what you actually want to describe is positive or negative deformation i.e. opening or closing of the cleft. Right? Please adapt the wording here.

**4.2 Crack Deformation Data** In general this paragraph is very descriptive and basically repeats what can be seen in figure 5. You can shorten it a bit and focus on/summarize the core results (there is deformation simultaneously to rain/snowmelt or without rain /snowmelt, temperature correlates with this and that…) This is easier to follow and the reader gets it faster.

**Figure 6:** Again, what is zero curtain here? Is it zero curtain at the surface or CTT??? But then snow cover and zero curtain period would occur simultaneously. There is a zero curtain during the snow melt and you couldn't separate these two periods as you did it in the figure … Completely unclear to me….

**Discussion**

**P9 l5 ff** Any thought why there is no spring zero curtain in the GT in 3 m depth? (but an autumn zero curtain?)

**P9 l11/12** again see Phillips et al 2016

**P9 l24:** Within the surface? At the surface or within the ground/rock mass!

**P9 l26:** refer to autumn zero curtains in 2 and 3m depth which are a proof for significant amounts of water in the rock mass!!

**P11 l15:** I am quite skeptical about the relevance of debuttressing in headwalls of glaciers. This is an often heard hypothesis which established more by repeating it again and again than by sound research

on it. You cited Keuschning et al but in this paper debuttressing is once more mentioned as important factor without giving any sound justification. The efficiency of debuttressing was shown for lateral slopes of valley glaciers but not for headwalls. Here we see erosion processes like plucking in the bergschrund causing oversteepening. This is a sign for glacial forces that rather act in the same direction as the critical rock slope deformation and not against the rock slope deformation. Rock masses obviously detached from the headwall as they were still covered by glacier ice, otherwise there would not appear an oversteep rock wall under the melting glacier. Rock masses still detach during deglaciation and after glaciation. Perhaps they detach more often than, as the atmospherically forced processes which you have measured in your nice dataset are more efficient than. But perhaps this is wrong and we are completely of the track! Perhaps the rock falls in freshly deglaciert areas are just an adaption process following oversteepening!? This is the big question that we should try to answer!

**P12 l14** See also Kenner *et al.* (2011) how observed the same at the summit wall of Gemsstock

**P12 l16** I think it is absolutely right that you emphasize shallow instabilities. But are you sure that erosion increased after deglaciation compared to the period during which it was ice covered? How do you know? Erosion during glacier coverage was obviously strong as well, as I said before: otherwise there would not appear an oversteep rock wall under the melting glacier. The only difference is that you can see the rock slides now and before they were invisible because they took place below the glacier line. I do not say that the one thing or the other is right or wrong but I consider it as an open question on which your paper could not give a satisfying answer so far.

Kenner R, Phillips M, Danioth C, Denier C, Zgraggen A. 2011. Investigation of rock and ice loss in a recently deglaciated mountain rock wall using terrestrial laser scanning: Gemsstock, Swiss Alps. *Cold Regions Science and Technology* 67: 157-164. doi: 10.1016/j.coldregions.2011.04.006
Phillips M, Haberkorn A, Draebing D, Krautblatter M, Rhyner H, Kenner R. 2016. Seasonally intermittent water flow through deep fractures in an Alpine Rock Ridge: Gemsstock, Central Swiss Alps. *Cold Regions Science and Technology* 125: 117-127. doi: S10.1016/j.coldregions.2016.02.010

---

## Referee Comment (RC2) · Anonymous Referee #2 · 29 May 2019

Dear Editor,

I have reviewed the paper "Fracture dynamics in an unstable, deglaciating headwall, Kitzsteinhorn, Austria" with great interest. The authors present a new dataset consisting of fracture displacement and temperature measurements over 2.5 years. This study observes reversible fracture displacement due to thermo-elastic strain of the rock mass, concludes that the percolation of water into fractures can result in irreversible fracture displacement and thereby supports the findings of a series of other studies in bedrock permafrost (e.g. Blikra & Christiansen, 2014, Weber et al., 2017, Draebing et al., 2017).

For me this manuscript presents an interesting study based on a new dataset, but it needs to clarify the methods used and increase the novelty in interpreting/discussing

the results before it can be published in The Cryosphere.

MAJOR POINTS

1) How does steep rock slopes differ from deglaciated headwalls? Without a direct comparison, it is difficult/critical to assign processes to deglaciation.

2) I find the term deformation for discontinuities or fractures/cracks confusing or problematic. I associate 'deformation' in rock mechanical contexts with a continuum, so a deforming fracture would be one that changes for instance shape from being planar to being curved. You are referring to movement of one side of the fracture with respect to the other one, while the fracture itself remains undeformed. I suggest using to use the term 'dislocation' for fractures (i.e. infinite deformation along a nominally flat fracture with very small aperture), and leave the term deformation for intact rock.

3) You focus on steep rock slope but gravitation is totally missing in interpretation and discussion.

4) You often mix results and discussion. I would suggest to clearly distinguish them.

5) Figure quality could overall be improved.

6) For me, the installation setup is not fully clear. I do not understand for certain what the two different crackmeters exactly measure. Therefore, it is difficult to fully review the results and discussion. Please also clarify the methods used.

7) Think about to refine the research questions including some novel idea/thoughts. The first two questions are mainly answered by several studies for the third one you do not have any evidence!

8) You often relativize your findings by statements like "... not be detected with the current measurement setup.", "... not been observed in this study." or similar.

MINOR POINTS P1L7f: The first sentence is hard to read. P1L19f: "Lower magnitude horizontal deformation occurs in autumn and early winter due to ice segregation."

[Figure]

Interactive
comment

is contradictory to the sentence on P10L25f P1L20: You state that "Irreversible fracture opening was not observed...". How do you know that cryogenic processes occur? Please take care on clearly formulated statements. P1L28: Add reference after "... at glacier cirques." P1L28: "Here, ..." - in general? Or where? P2L1f: "... rendering deglaciating headwalls particularly prone to rock slope failure." - is this link investigated? P2L3: Is this statement valid only for deglaciation triggered rock-slope instability or for rock-slope instability in general? P2L12: "... in rock slopes." - in general or deglaciated? P3L4ff: In this paragraph you mainly focus on monitoring but your intention is not to reduce your paper to a data set. P3L10-12: Are these research questions site specific or in general? P3L13ff: This paragraph shows the result. I would not put it in the introduction. P3L18: This section is much too short. I would extend it! And why Kitzsteinhorn? Is it representative? What are the coordinates and what is the elevation of the study site? P3L21f: Is the headwall now vertical or is the slope "only" 45°

Fig. 1: Where do you measure CTT? Where is the borehole (for the temperature measurements) located? Where is the weather station installed? (a) Scale not readable. North arrow not really readable. Coordinates would be nice. (c+d) Where are the crackmeters installed? (e+f) Where is top? Where are K1 and K2? P4L2: "... undergoing rapid deglaciation." - all slopes? Were all rock slopes at the Kitzsteinhorn glaciated in the near past?

P4L8: Which logger did you use? P4L11: Why do CDH and CDV not have the same resolution? Do you really mean resolution? Or accuracy? P4L17: "Temperature readings are taken every 10 minutes." ... and for the crackmeters? P4L19 "... and corrected for temperature variations." How? P4L21: Haberkorn adapted the method by Schmid to steep rock-slopes. Mention this! P4L21: And Haberkorn used this method based on near surface rock temperature. You are using air temperature. Can the method anyway be applied? P4L21: "... standard deviation of ..." over which period? 1 hour? 1 day? 1 week? P4L28f: You assume that all displacement occurs between the two blocks. What's happening at the outer ending of the blocks? No expansion there?

P4L30: There are four L's in Fig. 2. Please use L1, L2, H1 and H2. Eq. 2: L should be (H1-H2), or not? (H1 = height of Block 1; H2 = height of Block 2) P5L4: How many days a year do you stay at this condition? P5L4f: I can't follow this assumption. Why can you exclude ice segregation based on CTT data? At 2m depth, which is not below the block, the minimum temperature is >-8°C...

Fig. 2: What is below the blocks? What's at the interface below the blocks? You never mention gravitation, which is important in steep rock slopes. If you had a block with only friction on a slope, thermo-elastic strain would cause irreversible displacement. If cryostatic pressure opens a fracture, why would it close again?

Fig. 3: In this figure, it is not obvious how you get the zero curtain and snow cover periods as air temperature is fluctuating strongly. Probably from CTT in Fig. 4? If yes, why do you show air temperature in this figure? The precipitation is visualized for which periods (mm/hour or mm/day)? a) Why are there so many small gaps in precipitation and air temperature? a)+b) Add ticks for months. b) It's difficult to see the orange line on the gray area, e.g. the peak indicated with the black arrow. P5L32: "... indicating water percolation in the joint system." This is rather interpretation/discussion than result...

P6L3f: "... exceeded the contraction limit of the crackmeter so only minimum values for CDV were established." I do not understand. P6L4: "For most of the time, ..." And otherwise? When not? P6L5: "... negative correlation ..." Where? Have you done any correlation analysis? P6L9: Are these 0.2mm due to the thermo-elastic strain of the rock or a sensor artefact? P6L11f: Why? Is ice formation unlikely? P6L13: I don't understand. Fig. 6 shows the opposite! P6L16: Remove one "the". P6L16: Check the correct date format of copernicus. P6L17f: Very interesting!

Fig. 4: CTT in a) and b) the same? Why is the zero curtain not at 0°C? For better readability, I suggest having 3 subplots (1 temperature, 1 CDH, 1CDV) and may be combine with Fig. 7 (model output). In addition, I would a figure showing the temperaturedisplacement relation: Once measured data and once modelled data. Like this you see if you remove the reversible component due to thermo-elastic strain.

Fig. 5: It's hard to compare the subplots as a)-c) have different ranges for the y-axes (e.g. for temperature, a = 15°, b=20°, c=10+). Is the weather station representative (close enough)? Topography is relevant: gully vs. spur like feature.

Fig. 6: It took me a long time to realize what * and ** are indicating. Was the rock-slope already covered by snow in September 2017? It looks like in b) but Fig. 4 says no? It is rather difficult to track the lines/points over time and to link a) and b).

P7L16: Why? How do you know? Fig. 8 shows a big range of alpha below -10°C.. P7L19: "... good agreement ..." correlation analysis is needed! P7L22: Pleas show the residual displacement!

Fig. 7: I would like to see the residual displacement. a) Your model shows a fracture closing in winter 2017/2018, but your data not... What's the horizontal yellow line in July 2017? b) You have less than 0.5 summer with vertical displacement data and there is almost no agreement with the model even in winter (2016/2017). Are the results still representative? a)+b) add ticks.

P8L2: "Data Quality ..." is not explained. P8L3-L11: This paragraph rather describes assumptions and limitations. P8L13: "... significant negative correlation ..." I haven't seen any correlation analysis! P8L13f: Only for certain time periods... P8L17: Give numbers! Yours in comparison to others. P8L20: "... not show significant ..." Analysis? P8L21f: Why are the boundaries exact at 0? The upper boundary in Fig. 8b is marked wrongly, it does not go up to 5E-5.

Fig. 8: Autumn cooling and spring warming can't be distinguished. Could you indicate these periods with different colors? Why is ice segregation in a) and b) indicated differently?

P9L10: "... causing the observed rapid contraction at the end of the zero curtain period." I do not understand this statement. P9L14f: Do you have any evidence in the data? Or is it a conceptual thought? P9L17: "... no ... mechanical reaction ..." Why? And again significant... P9L20ff: This statement/interpretation is rather vague... P9L24: "... freezing of rainwater ..." Is this scenario often the case? P10L2ff: Hard to follow. P10L8ff: Temperature in Fig. 8 refers to 10day-means. However, in reality there is much more fluctuation and the temperature does not stay in the range mentioned... P10L22: You totally ignore the influence of gravity! P10L25: "... growth on a seasonal scale." Fracture is NOT growing/opening (see your figures), it closes again! P10L25f: Are you really able to observe ice segregation? What's the evidence? If yes, how do you know the depth? P11L1: ... or CTT is just not representative?! P11L9ff: This is rather a "review" and not really based on or linked to your observations. P11L24: "... low magnitude rock detachments." Have you monitored/observed them? P11L24: "The contribution of an individual fracture to the stability of the entire headwall may thus seem negligible." Assumption? Result? Interpretation? P11L28: "... than a centime-ter." You don't show any data supporting this! P11L29f: You don't show this in Fig. 2. Why? P11L31: What's about gravity? P11L32f: "Our data shows that temperature in combination with moisture availability is the critical factor determining enhanced frac-ture deformation on diurnal to seasonal scales (Matsuoka, 2008)." Does your data or Matsuoka (2008) show it? P12L3: Repetition... P12L12-14: "A related ... in prepara-tion)." I would move this to the introduction/motivation. P12L15: Really? Does it? You do not show/compare fracture that was exposed longer time. Without such a compari-son, your statement has no evidence is therefore rather critical. P12L28: In my opinion, you can't conclude this statement...

I hope that these comments help for improving the presentation of this work.

---

## Referee Comment (RC3) · Anonymous Referee #3 · 5 Jun 2019

Ewald et al. present results of 2.5 year on-site monitoring of crack dynamics in an unstable alpine rockwall underlain by permafrost, including a valuable data set that indicates interesting correlations between 2D crack deformation and thermal conditions down to 5 m depth. Since on-site observations at high-risk rockwalls are extremely limited, the data can potentially contribute to geomorphology and engineering geology in cold regions.

However, the paper includes a number of fundamental problems mainly arising from the lack of originality, missing detailed information on monitoring, inadequate modelling, disregard of heat conduction and unclear interpretations. Observed large vertical movements unaccompanied by horizontal movements during zero-curtain periods are interesting, but interpretation seems unsuccessful. Also, the absence of irreversible

crack opening does not prove the contribution of the observed movements to the instability of the monitored block as preparatory processes. The modelling is confusing, since it does not successfully separate thermo-elastic components from other (e.g. cryogenic) components (cf. Weber et al., 2017) and, furthermore, it discusses temporal variability of a constant (thermal expansion coefficient of rock). As a result, plausible conclusions have not yet been reached. In conclusion, the paper is not acceptable in its present form, but it may become acceptable when carefully and thoroughly revised.

The major issues are listed below: the third one is most serious.

1. What is the novelty of this paper? Whereas the paper presents data from a single crack for 2.5 years, Hasler et al. (2012) and Weber et al. (2017) have already presented data on 2D deformation of several cracks, suggesting several types of triggers. Draebing et al. (2017) presented data on horizontal deformation, temperature and water level of three cracks facing different aspects, discussing detailed thermo-hydrological conditions of the cracks. The analysis in this paper mainly follows Draebing et al. without any advance/improvement. Overall, what are the strong points of this paper? Perhaps the borehole temperatures may help discussion of the correlation between thermal condition and crack deformation at depth?

2. The methodology should be more clearly illustrated. The photographs (Fig.1e,f) only display protectors, but do not show the installation of crackmeters. I suspect that the crackmeters were installed like Fig.R1 (see supplement). If it is correct, the two components, CDH and CDV, have to be interrelated. In addition, information on CTT and boreholes are lacking. Is CTT measured at the entrance of the crack or in the casing of crackmeter? Please show the location of the borehole in Fig.1 – on the rockface or on the top station? Detailed illustration of these features is necessary for plausible interpretations.

3. Data analysis is unreasonable or questionable in the following points. First, Equation 1 assumes that the whole lengths of two blocks (B1 and B2 in Fig.R2) enclosing

the observed crack contribute to the horizontal deformation (CDH) of the crack, but in reality, only half of both blocks contribute to it while the other halves contribute to the next cracks (see Fig.R2). Thus, I believe that the contributing length is L rather than 2L. Similarly, CDV is affected by outward (or inward) movements of the both blocks, instead of the authors' approximation that CDV is taken to represent the deformation of one single block (the last sentence of page 4). In this case, when the rockwall is heated, both B1 and B2 blocks may expand outward (see Fig.R2), cancelling the movement of each block; as a result, CDV represents the deformation derived from the difference in the heights of the two blocks. Second, Fig.8 shows temporal changes in the thermal expansion coefficient, but this is unrealistic. The coefficient must be constant for each rock (also cannot be negative), so the assumption in the modelling is likely to be wrong. Third, I suspect that the model applies the parameters derived from on-site monitoring to the results of on-site monitoring. The argument seems to constitute circular reasoning, and the agreement between the monitoring and model should be a natural consequence? Fourth, the modelling assumes 'fracture deformation at CTT below -10°C is governed exclusively by thermo-mechanical controls', but deformation at positive CTT (e.g. values during the midsummer) are also largely governed by thermo-mechanical control (except for some events associated with rainwater) so that it can also be used 'to unravel thermo-mechanical from cryogenic fracture deformation'. In fact, however, modelled deformation shows five times wider daily fluctuations in summer than measured deformation (Fig.7a), which indicates the invalidity of the assumption. Perhaps a different coefficient should be used for data above 0°C?

4. Three kinds of thermal windows (TE, FT, IS) are proposed in Fig.8, but I wonder whether the analysis is appropriate or not. This is because FT and IS should be defined by the temperature at a depth where deformation actually occurs, instead of the surface (crack-top) temperature. When heat conduction (i.e. time lag in cooling/warming) is taken into account, the former temperature could be significantly higher than the latter during intensive cooling. Are there any supporting data or reason for substituting the former for the latter?

5. The observed vertical expansion during the spring zero-curtain period is very interesting, but needs more careful interpretation, preferably with illustration. The authors attributed the increase in CDV to refreezing of rain-/melt-water. This is a possible explanation for 'horizontal' expansion, but why CDH did not increase. The vertical movement might be explained by frost heaving of the upslope block (B1) at the bottom, followed by settlement of B1 upon thawing (see Fig.R3), but is it realistic? However, this interpretation still cannot explain the absence of change in CDH. Reconsideration is necessary.

Specific comments (Page/Line) P2/L17: I suspect that Wegmann and Gudmundsen (1999) do not attribute seasonal crack movements to thermo-mechanical process but to seasonal freezing-thawing (volumetric expansion or ice segregation). P3/L5: ...immediately upslope of the... P4/L7: Add information on the data logger. P4/L11: Why are resolutions different between two crackmeters? P4/L12: Show the location of thermistor (for CTT) in Fig.1 (or 2). P4/L16: Show the location of the borehole in Fig.1. P4/L30: The definitions of deltaCD and deltaCTT are unclear. What is the 10-day mean crack deformation: the difference between the maximum width and minimum width in 10 days, or anything else? DeltaCTT appears to represent a change in crack-top temperature, but indeed it is defined by the 10-day mean CTT value (not a change but a single temperature). P5/L20: Use consistent units: ...depth of 2 m, 3 m and 5 m... P5/L22: Define RT300. P5/L25: 'with no obvious correlation to snow cover thickness': But does the snow cover thickness represent the value at the observed crack? P7/L15: I recommend to draw a graph showing the derivation of alpha values. Fig.1: Do the red face and blue face in (d) represent K1 and K2, respectively? Fig.3: How are the zero-curtain periods determined? – derived from CTT values in Fig.4? Fig.4: Why did data gap occur in spring 2017 – not mentioned in the caption/text.

All of the references cited here are listed in the original text.

Please also note the supplement to this comment:
https://www.the-cryosphere-discuss.net/tc-2019-42/tc-2019-42-RC3-supplement.pdf

**Supplement:**

*Monitoring condition (assumed)*

**When heated**

L1/2 contributes to closure of K2

L2/2 contributes to closure of K2

**K2** joint

H1=3.0 m

Whole H1 contributes to heaving of B1

**B1** block

CDH

CDV

H1=3.0 m

CDH

CDV

L1=7.0 m

5 cm

L1=7.0 m

5 cm

Whole H2 contributes to heaving of B1

H2=2.6 m

**B2** block

H2=2.6 m

L2=7.0 m

L2=7.0 m

*Fig. R1*

*Fig. R2*

$\Delta CDH = -(L1/2 + L2/2)\ \alpha H\ \Delta CTT$
$\Delta CDV = -(H1-H2)\ \alpha V\ \Delta CTT$

[Figure]

*Fig. R3*

Heave?

Settlement?

Meltwater

B1

B2

B1

B2

B1

B2

Refreezing?

Frost heave of B1?

Thawing?

*Why does vertical extension-contraction occur without horizontal movement in spring?*

---

## Short Comment (SC1) · Andreas Ewald et al. · 7 Jun 2019

Fracture dynamics in an unstable, deglaciating headwall, Kitzsteinhorn, Austria

Andreas Ewald et al.

The manuscript by Andreas Ewald et al. discusses experiments and their respective results regarding kinematic observations made in a steep, high-alpine rockwall (per-mafrost) in the European Alps. In general this is interesting and timely work. Research in this area is appreciated by many although similar work exists/is performed only by a (very) small community. As a result the body of knowledge and related work is com-pact and many open questions exist. And exactly in this respect I feel that the present manuscript lacks focus and tries to solve (elude to) too many problems at once. Many of

the claims made are not substantiated by evidence (observation/models) and in parts are contradicting. As such i suggest to limit the manuscript to relate only to processes that are either known and defined in related work or are clearly visible in the data and analysis provided.

The title suggests that an "unstable, deglaciating headwall" is discussed in this paper. While i have no doubts that (significant) glacier retreat also takes place at Kitzstein-horn, no evidence (retreat rates, references, differential DEM, photos) are given to back up the "deglaciating". W.r.t. to the term "unstable" the paper later states that "irre-versible fracture opening was not observed". Additionally there is no further evidence given (rockfall observations, large-scale kinematic observations, debris) that back up the term "unstable". Similarly the term "headwall retreat" should be backed up as well. How much? What is known here? What is observed? Given the collocation with in-frastructure (cable car, ski resort) long term evidence should be available apart from regional spatial data and references.

With respect to the data presented there are some issues that should be fixed: The temperature "Crack Top" is not properly described ("Crack top temperatures may not represent the entire fracture"). Where is it measured and what does this represent exactly? The air temperature and snow height that is in large parts used for process analysis is measured on the glacier, yet your rock wall is north facing above the glacier (up to 100m altitide). How do you correct this air temperature/snow height to reflect conditions inside the steep north-facing rock wall? Your precipitation measurement is from a station ~500-600m lower in altitude and ~2km away. This gives you an impression of the regional precipitation (sum), but in an alpine setting it is doubtful if it really captures event-by-event details w.r.t. precipitation, especially for the steep rockwall environment and strong (warm) summer liquid precipitation you are targeting. A co-located precipitation sensor would be highly beneficial here. The model developed here (section 4.3) lacks detail. In it's current state, the model cannot be reproduced.

Last I want to comment that you regularly relate to "randkluft systems" and to people

knowing your project (history) it is known that you are actively exploring/instrumenting also the perimeter of the Kitzsteinhorn rock walls below the glacier surface but this paper does not show any evidence from below the glacier surface. Therefore any connect to processes specific to this "randkluft" regime, e.g. last sentence of the abstract, middle section of section 5.3 are highly speculative at the best. Similarly, speculative statements ("other mechanisms may affect fracture dynamics" "may lead to...") do not offer any sound interpretation of what is/can be observed in this case study. The general impression that this manuscript is not quite mature yet is further exacerbated by the fact that figure captions and figures are not located together, which makes the manuscript rather hard to decipher.

---

## Author Comment (AC1) · 18 Nov 2019

We thank Robert Kenner for his insightful and constructive comments. The expressed criticism is substantial and points to a lack of novelty and originality in the submitted manuscript. We therefore consider a complete revision of our manuscript. The intended new manuscript will no longer focus on recent deglaciation, instead we will explore the relation between fracture kinematics and active layer dynamics in steep, frozen rockwalls. For this purpose we will now concentrate on regression analyses between data from the described crackmeter station and an adjacent permafrost borehole. We consider the immediate vicinity of a deep borehole and a crackmeter station a novel measurement setup that has the potential to advance the current knowledge on kinematics in steep bedrock permafrost. To increase the significance of the new

analysis we will expand the time series to four years (2016-2019) as opposed to 2.5 years in the current manuscript. To model thermo-elastic deformation we will resort to the linear regression model published by Weber et al. (2017), and will no longer derive the thermo-elastic deformation component from cracktop temperatures below -10 °C, which has been criticized by all reviewers. To identify potential discrepancies between measured and modeled fracture kinematics we intend to implement state-of-the-art ice segregation models driven by borehole temperature data. Below, we respond to the major concerns/points of criticism of Robert Kenner only. Facing significant changes of the revised manuscript we assume that most minor comments will no longer apply, therefore we will not comment on here. More general comments of this section will be considered in the revision process.

REVIEWER COMMENT: [. . .] Furthermore, I have the feeling that the significance of the paper could be clearly increased if the authors establish a link to the really central question: What is role of deglaciation?

REPLY: Former glacial occupation led to a change in slope form. The schrundline dissects the headwall into an upper, flatter (45°) and a lower, steeper (90°) part. Headwall retreat, presumably driven by periglacial weathering in the randkluft system (Sanders et al., 2012), likely created the steep, lower section. With the onset of deglaciation, atmospheric forcing is hypothesised to significantly contribute to the widening of pre-existing fractures and thus to a destabilisation of the deglaciated rock slope. (We will add a sketch with a cross section of the headwall from the summit station to the recent glacier surface to better illustrate the whole setting.) As we are not able to compare the fracture kinematics of this paraglacial setting with a periglacial one, we decided to rather focus on active layer dynamics and its mechanical response.

REVIEWER COMMENT: [...] However, to stick out and become a really important paper the authors have to interpret their results with respect to glaciation. Are the observed processes particularly related to a deglaciered rock wall or can I observe the same in the upper parts of the rock wall which was never covered by glacier ice?

[Figure]

REPLY: What we observe in our data may also be observed in other (never glacier covered but permafrost affected) parts of the rock wall. Freeze-thaw action as well as ice segregation are common processes acting to destabilise permafrost affected rock walls. The question is whether these processes are particularly efficient in a rather unfavourable geotechnical setting (vertical rock wall with steeply outdipping fractures) preconditioned by glacial erosion. However, again, we will shift the focus away from deglaciation.

REVIEWER COMMENT: What is difference to rock wall kinematics below the glacier line or in the Bergschrund? Is the erosion faster above or below the Bergschrund? In particular the last question is considered by too many authors as obvious but the answer is not obvious at all.

REPLY: As we present a point data set from a location above the bergschrund, links to the bergschrund environment are more likely to be seen as an outlook. A comparison of the efficiency of erosion above or below the glacier line is beyond the limits of this study but we are working on that.

REVIEWER COMMENT: Abstract: I miss some information about permafrost presence or absence. This is an important factor that should be mentioned in the Abstract. Searching "permafrost" in the rest of the document gave no clear answer on this question neither.

REPLY: We added detailed information about the ground thermal regime based on the borehole temperature data to address the permafrost occurrence in our study site.

REVIEWER COMMENT: Introduction: P1 l29: This is an interesting (and correct) description but do you have some explanations or references why these headwalls are often oversteepened? I think there is a term for it called schrundlines. See e.g. sanders et al 2012 (cited by you). These schrundlines have obviously developed during glaciation and not after glaciation or during deglaciation. Any thoughts on that?

REPLY: Schrundlines, as the name says, mark the location of the (former) bergschrund which divides the glacier body into a dynamic, presumably warm-based glacier and an upper, cold-based hanging glacier (citation). The type of glacier cover is of crucial importance, since the static, upper part may prevent erosion whereas the lower, dynamic part may favour plucking, abrasion and periglacial weathering inside the bergschrund. The discrepancy of favoured and prevented erosion may be expressed by this distinct "knickpoint"/schrundline. The resulting form is preserved by high rock mass strength (permafrost!) and destabilised by permafrost thaw following deglaciation. To better demonstrate the link between the form of the headwall, the importance of the schrundline and the location of our instrumentation, we will add a sketch with a cross-section of the headwall similar to figure 10 in Keuschnig et al., 2017.

REVIEWER COMMENT: P2 l6ff: I would distinguish these processes more carefully. Debuttressing is probably not a driving factor at headwalls but is more related to lateral (valley) slopes of tongue shaped glaciers. Oversteepening occurs at headwalls as well as on lateral slopes but the type of glacial erosion is very different between both locations. (Perhaps rather a type of plucking behind the bergschrund!? See again sanders et al.) If you want to focus on headwalls, I think it is important to go more into detail here.

REPLY: Oversteepening due to glacial erosion may dominate over debuttressing as preconditioning/ preparatory factor but the toe of a headwall may still be buttressed by the cirque glacier (ice-rock contact). However, as we decided to readjust our research focus we feel that this discussion is beyond the scope of this publication.

REVIEWER COMMENT: Furthermore, I feel it is a pity that all rock slope failures taking place in the vicinity of a glacier are lumped together by most of the studies in this field. Oversteepening is obviously a result of glacier erosion and not of glacier retreat. Rock slope instabilities caused by oversteepening are thus just secondarily or not related to glacier retreat. This is probably different for rock slope failures whose kinematic was not related to oversteepening or which were previously covered by glacier ice in large

parts (especially the discontinuities) as it is the case at your study site. Just recently we observed a rock fall at Flüela Wisshorn in Switzerland where more than 250'000 m3 collapsed in an old glacier cirque (you can google some pictures of it). This cubature was never covered by glacier ice, not even during the last cold period. However, the cubature was kinematically free, as the release plane (dip slope) cropped out below the cubature already before the event. This was because the lower end of the cubature was built by a terrain step which was part of a distinct schrundline. In such a case it makes absolutely no sense for me to talk about a slope failure related to glacier retreat. This instability originates most likely from glacial erosion during the last glacial maximum and collapsed now, several decades after the LIA glaciation and several thousands of years after its initiation. I claim it would have collapsed as well if the LIA glacier below the cubature would still have been present. Maybe you can consider those differences somewhere in the introduction and perhaps also in the discussion

REPLY: As you mentioned, the locations of rock slope failures observed in the headwall at our study site were formerly covered by cold-based glacier ice (above the schrund-line). Consequently RSFs may be a result of glacier retreat. This discussion is outside the scope of this study.

REVIEWER COMMENT: Data Acquisition P4 l16: Why just in 3 depth if the borehole is 30 m deep? Why not deeper and why not closer to the surface (to track freezing fronts)?

REPLY: As mentioned above, we will add borehole temperature data from thermistors at least up to a depth of 10 meters.

REVIEWER COMMENT: Data Analysis P5l5: "(i) fracture deformation by ice segregation is of minor importance" Can you give an explanation for this? CTT are a weak indicator for the temperature profiles in greater depth aren't they?

REPLY: To model thermo-elastic deformation we will resort to the linear regression model published by Weber et al. (2017), and will no longer derive the thermo-elastic

deformation component from cracktop temperatures below -10 °C, which has been criticized by all reviewers. By carrying out correlation analyses between fracture displacements and rock temperatures from the borehole at different depths we will be able to see at which temperatures in which depth significant fracture displacements occur.

REVIEWER COMMENT: 4 Results

Figure 3: The rock temperatures show distinct zero curtains in Autumn but not in spring. This is interesting and important. Is this somewhere discussed? There must be a lot of water somewhere in the rock that caused the long autumn zero curtain by freezing...? But why is there no ZC in spring? Where has the ice gone!? Or has the water percolated away in autumn without freezing??

REPLY: Meltwater may percolate away in spring due to active layer thaw from bottom to the top (see Keuschnig et al. 2017). Surface freezing in autumn may lead to huge amounts of perched water within the rock mass. We will include these issues in the discussion.

REVIEWER COMMENT: P5 l22 Anomalies during zero curtain period? You are talking about ZC at which depth? Increase of 1K during zero curtain at 3m depth means +1°C what is not possible as active layer depth was 3m in 2016!? Along with shorter zero curtain period... Where? When? Unclear! Section is hard to follow, try to formulate more precise.

REPLY: Zero curtain is detected at the crack top. We will formulate the chapter more precisely and include a sketch to better illustrate the setup.

REVIEWER COMMENT: Figure 6: Again, what is zero curtain here? Is it zero curtain at the surface or CTT??? But then snow cover and zero curtain period would occur simultaneously. There is a zero curtain during the snow melt and you couldn't separate these two periods as you did it in the figure ... Completely unclear to me....

REPLY: We tried to clarify the use of the term "zero curtain" in the manuscript. In

this figure the term may be misleading. Classes in figure 6 are related to temperature ranges and not solely to snow cover. However, we translate positive temperature to snow free conditions and negative temperatures to snow covered conditions. In this case the period of ZC represents a phase where snow cover is still present, but in a melting phase. We will modify this figure accordingly to avoid this overlap.

REVIEWER COMMENT: Discussion

P11 l15: I am quite skeptical about the relevance of debuttressing in headwalls of glaciers. This is an often heard hypothesis which established more by repeating it again and again than by sound research on it. You cited Keuschning et al but in this paper debuttressing is once more mentioned as important factor without giving any sound justification. The efficiency of debuttressing was shown for lateral slopes of valley glaciers but not for headwalls. Here we see erosion processes like plucking in the bergschrund causing oversteepening.

REPLY: Do we really see plucking in the bergschrund? Can you provide any reference for this? Our observations do not reveal indication on plucking.

REVIEWER COMMENT: This is a sign for glacial forces that rather act in the same direction as the critical rock slope deformation and not against the rock slope deformation. Rock masses obviously detached from the headwall as they were still covered by glacier ice, otherwise there would not appear an oversteep rock wall under the melting glacier. Rock masses still detach during deglaciation and after glaciation. Perhaps they detach more often than, as the atmospherically forced processes which you have measured in your nice dataset are more efficient than. But perhaps this is wrong and we are completely of the track! Perhaps the rock falls in freshly deglaciert areas are just an adaption process following oversteepening!? This is the big question that we should try to answer!

REPLY: Since we are not able to answer this important question with our data set we are not going to enhance the discussion on whether glacial debuttressing is a preparatory factor or not. However we are recently trying to tackle these questions in our further research activities. Since we will not stress the deglaciation aspects in the revised version of the manuscript, this will no longer apply here.

REVIEWER COMMENT: P12 l16 I think it is absolutely right that you emphasize shallow instabilities. But are you sure that erosion increased after deglaciation compared to the period during which it was ice covered? How do you know? Erosion during glacier coverage was obviously strong as well, as I said before: otherwise there would not appear an oversteep rock wall under the melting glacier. The only difference is that you can see the rock slides now and before they were invisible because they took place below the glacier line. I do not say that the one thing or the other is right or wrong but I consider it as an open question on which your paper could not give a satisfying answer so far.

REPLY: Here again, it is to add that the part of the headwall, which we instrumented, has been covered by cold-based glacier ice which presumably prevented extensive fracture displacements. The location of our crackmeter is not within the randkluft, therefore other processes may apply.

---

## Author Comment (AC2) · 18 Nov 2019

We thank Anonymous Referee #2 for his/her insightful and constructive comments. The expressed criticism is substantial and points to a lack of novelty and originality in the submitted manuscript. We therefore consider a complete revision of our manuscript. The intended new manuscript will no longer focus on recent deglaciation, instead we will explore the relation between fracture kinematics and active layer dynamics in steep, frozen rockwalls. For this purpose we will now concentrate on regression analyses between data from the described crackmeter station and an adjacent permafrost borehole. We consider the immediate vicinity of a deep borehole and a crackmeter station a novel measurement setup that has the potential to advance the current knowledge on kinematics in steep bedrock permafrost. To increase the significance of the new

analysis we will expand the time series to four years (2016-2019) as opposed to 2.5 years in the current manuscript. To model thermo-elastic deformation we will resort to the linear regression model published by Weber et al. (2017), and will no longer derive the thermo-elastic deformation component from cracktop temperatures below -10 °C, which has been criticized by all reviewers. To identify potential discrepancies between measured and modeled fracture kinematics we intend to implement state-of-the-art ice segregation models driven by borehole temperature data. Below, we respond to the major concerns/points of criticism of Anonymous Referee #2 only. Facing significant changes of the revised manuscript we assume that most minor comments will no longer apply, therefore we will not comment on here. More general comments of this section will be considered in the revision process.

REVIEWER COMMENT: MAJOR POINTS

1) How does steep rock slopes differ from deglaciated headwalls? Without a direct comparison, it is difficult/critical to assign processes to deglaciation.

REPLY: We agree. The intended new manuscript will account for this concern by focusing less on the impact of deglaciation, which cannot be quantified adequately with our measurement setup - and will instead focus on the correlation between fracture kinematics and active layer dynamics.

REVIEWER COMMENT: 2) I find the term deformation for discontinuities or fractures/cracks confusing or problematic. I associate 'deformation' in rock mechanical contexts with a continuum, so a deforming fracture would be one that changes for instance shape from being planar to being curved. You are referring to movement of one side of the fracture with respect to the other one, while the fracture itself remains undeformed. I suggest using to use the term 'dislocation' for fractures (i.e. infinite deformation along a nominally flat fracture with very small aperture), and leave the term deformation for intact rock.

REPLY: We will follow your suggestion and use the term fracture 'dislocation'.

REVIEWER COMMENT: 3) You focus on steep rock slope but gravitation is totally missing in interpretation and discussion.

REPLY: We will put a stronger focus on the integration of gravitation in the new intended manuscript.

REVIEWER COMMENT: 4) You often mix results and discussion. I would suggest to clearly distinguish them.

REPLY: We will revise the document and separate results and discussion parts more thoroughly.

REVIEWER COMMENT: 5) Figure quality could overall be improved.

REPLY: We will cross-check figure quality and improve the quality if required.

REVIEWER COMMENT: 6) For me, the installation setup is not fully clear. I do not understand for certain what the two different crackmeters exactly measure. Therefore, it is difficult to fully review the results and discussion. Please also clarify the methods used.

REPLY: We will add another sketch depicting the installation setup to better illustrate the operation of the instruments used. We also look through the methods section to be more precise here.

REVIEWER COMMENT: 7) Think about to refine the research questions including some novel idea/thoughts. The first two questions are mainly answered by several studies for the third one you do not have any evidence!

REPLY: Since we decided to include the borehole temperature data into our analysis new research questions will arise. One new research questions will be: How do temperature changes in the subsurface affect crack behaviour. Further, we intend to include frost cracking models to identify identify and explain potential deviations between measured and modeled fracture dislocation.
REVIEWER COMMENT: 8) You often relativize your findings by statements like "... not be detected with the current measurement setup.", "... not been observed in this study." or similar.

REPLY: We will revise our manuscript and change the language to avoid statement on findings, which we cannot prove.

---

## Author Comment (AC3) · 18 Nov 2019

We thank Anonymous Referee #3 for his/her detailed, insightful and constructive comments. The expressed criticism is substantial and points to a lack of novelty and originality in the submitted manuscript. We therefore consider a complete revision of our manuscript. The intended new manuscript will no longer focus on recent deglaciation, instead we will explore the relation between fracture kinematics and active layer dynamics in steep, frozen rockwalls. For this purpose we will now concentrate on regression analyses between data from the described crackmeter station and an adjacent permafrost borehole. We consider the immediate vicinity of a deep borehole and a crackmeter station a novel measurement setup that has the potential to advance the current knowledge on kinematics in steep bedrock permafrost. To increase the significance of the new analysis we will expand the time series to four years (2016-2019) as opposed to 2.5 years in the current manuscript. To model thermo-elastic deformation we will resort to the linear regression model published by Weber et al. (2017), and will no longer derive the thermo-elastic deformation component from cracktop temperatures below -10 °C, which has been criticized by all reviewers. To identify potential discrepancies between measured and modeled fracture kinematics we intend to implement state-of-the-art ice segregation models driven by borehole temperature data. Below, we respond to the major concerns/points of criticism of Anonymous Referee #2 only. Facing significant changes of the revised manuscript we assume that most minor comments will no longer apply, therefore we will not comment on here. More general comments of this section will be considered in the revision process.

REVIEWER COMMENT: MAJOR POINTS

1. What is the novelty of this paper? Whereas the paper presents data from a single crack for 2.5 years, Hasler et al. (2012) and Weber et al. (2017) have already presented data on 2D deformation of several cracks, suggesting several types of triggers. Draebing et al. (2017) presented data on horizontal deformation, temperature and water level of three cracks facing different aspects, discussing detailed thermo-hydrological conditions of the cracks. The analysis in this paper mainly follows Draebing et al. without any advance/improvement. Overall, what are the strong points of this paper? Perhaps the borehole temperatures may help discussion of the correlation between thermal condition and crack deformation at depth?

REPLY: Excellent point. We consider a revision of the present manuscript that will focus on the relationship between fracture deformation and borehole temperature data. Furthermore, we intend to use the borehole data to run a frost-cracking model.

REVIEWER COMMENT: 2. The methodology should be more clearly illustrated. The photographs (Fig.1e,f) only display protectors, but do not show the installation of crackmeters. I suspect that the crackmeters were installed like Fig.R1 (see supplement). If

it is correct, the two components, CDH and CDV, have to be interrelated. In addition, information on CTT and boreholes are lacking. Is CTT measured at the entrance of the crack or in the casing of crackmeter? Please show the location of the borehole in Fig.1 – on the rockface or on the top station? Detailed illustration of these features is necessary for plausible interpretations.

REPLY: We will provide an improved illustration that accounts for your suggestions.

REVIEWER COMMENT: 3. Data analysis is unreasonable or questionable in the following points. First, Equation 1 assumes that the whole lengths of two blocks (B1 and B2 in Fig.R2) enclosing the observed crack contribute to the horizontal deformation (CDH) of the crack, but in reality, only half of both blocks contribute to it while the other halves contribute to the next cracks (see Fig.R2). Thus, I believe that the contributing length is L rather than 2L. Similarly, CDV is affected by outward (or inward) movements of the both blocks, instead of the authors' approximation that CDV is taken to represent the deformation of one single block (the last sentence of page 4). In this case, when the rockwall is heated, both B1 and B2 blocks may expand outward (see Fig.R2), cancelling the movement of each block; as a result, CDV represents the deformation derived from the difference in the heights of the two blocks. Second, Fig.8 shows temporal changes in the thermal expansion coefficient, but this is unrealistic. The coefficient must be constant for each rock (also cannot be negative), so the assumption in the modelling is likely to be wrong. Third, I suspect that the model applies the parameters derived from on-site monitoring to the results of on-site monitoring. The argument seems to constitute circular reasoning, and the agreement between the monitoring and model should be a natural consequence? Fourth, the modelling assumes 'fracture deformation at CTT below -10C is governed exclusively by thermo-mechanical controls', but deformation at positive CTT (e.g. values during the midsummer) are also largely governed by thermo-mechanical control (except for some events associated with rainwater) so that it can also be used 'to unravel thermo-mechanical from cryogenic fracture deformation'. In fact, however, modelled deformation shows five times wider daily fluctuations

in summer than measured deformation (Fig.7a), which indicates the invalidity of the assumption. Perhaps a different coefficient should be used for data above 0C?

REPLY: These comments will no longer apply since we will use a different modelling approach. We consider the comment on L for the CDH model. Concerning CDV we acknowledge the comment and agree that this measurement is critical and hard to explain. Also with respect to the available data and large gaps in our measurements, we promise to be more careful when interpreting the CDV results.

REVIEWER COMMENT: 4. Three kinds of thermal windows (TE, FT, IS) are proposed in Fig.8, but I wonder whether the analysis is appropriate or not. This is because FT and IS should be defined by the temperature at a depth where deformation actually occurs, instead of the surface (crack-top) temperature. When heat conduction (i.e. time lag in cooling/warming) is taken into account, the former temperature could be significantly higher than the latter during intensive cooling. Are there any supporting data or reason for substituting the former for the latter?

REPLY: This is a great point. In the intended new manuscript we will refrain from using cracktop temperature as an exclusive indicator of FT and IS. We will instead analyze borehole data (maximum depth 30 m) and will furthermore use the linear regression model introduced by Weber et al. (2017) to model thermo-elastic (reversible) deformation.

REVIEWER COMMENT: 5. The observed vertical expansion during the spring zero-curtain period is very interesting, but needs more careful interpretation, preferably with illustration. The authors attributed the increase in CDV to refreezing of rain-/melt-water. This is a possible explanation for 'horizontal' expansion, but why CDH did not increase. The vertical movement might be explained by frost heaving of the upslope block (B1) at the bottom, followed by settlement of B1 upon thawing (see Fig.R3), but is it realistic? However, this interpretation still cannot explain the absence of change in CDH. Reconsideration is necessary.

REPLY: We will provide another figure that better explains our intentions. During the described period we found an increase in CDV that was not accompanied by a change in CDH. We therefore concluded that the increase in CDV can only be explained by expansion in the uppermost decimeters of the upper block - which are not covered by the horizontal crackmeter (CDH).

---

## Author Comment (AC4) · 18 Nov 2019

We thank Jan Beutel for his insightful and constructive comments. The expressed criticism is substantial and points to a lack of novelty and originality in the submitted manuscript. We therefore consider a complete revision of our manuscript. The intended new manuscript will no longer focus on recent deglaciation, instead we will explore the relation between fracture kinematics and active layer dynamics in steep, frozen rockwalls. For this purpose we will now concentrate on regression analyses between data from the described crackmeter station and an adjacent permafrost borehole. We consider the immediate vicinity of a deep borehole and a crackmeter station a novel measurement setup that has the potential to advance the current knowledge on kinematics in steep bedrock permafrost. To increase the significance of the new

analysis we will expand the time series to four years (2016-2019) as opposed to 2.5 years in the current manuscript. To model thermo-elastic deformation we will resort to the linear regression model published by Weber et al. (2017), and will no longer derive the thermo-elastic deformation component from cracktop temperatures below -10 °C, which has been criticized by all reviewers. To identify potential discrepancies between measured and modeled fracture kinematics we intend to implement state-of-the-art ice segregation models driven by borehole temperature data.

COMMENT: The manuscript by Andreas Ewald et al. discusses experiments and their respective results regarding kinematic observations made in a steep, high-alpine rock-wall (permafrost) in the European Alps. In general this is interesting and timely work. Research in this area is appreciated by many although similar work exists/is performed only by a (very) small community. As a result the body of knowledge and related work is compact and many open questions exist. And exactly in this respect I feel that the present manuscript lacks focus and tries to solve (elude to) too many problems at once. Many of the claims made are not substantiated by evidence (observation/models) and in parts are contradicting. As such i suggest to limit the manuscript to relate only to processes that are either known and defined in related work or are clearly visible in the data and analysis provided.

REPLY: Thank you for this constructive comment. We will consider a more thorough focus of our analysis and discussion for the revised manuscript version.

COMMENT: The title suggests that an "unstable, deglaciating headwall" is discussed in this paper. While i have no doubts that (significant) glacier retreat also takes place at Kitzsteinhorn, no evidence (retreat rates, references, differential DEM, photos) are given to back up the "deglaciating". W.r.t. to the term "unstable" the paper later states that "irreversible fracture opening was not observed". Additionally there is no further evidence given (rockfall observations, large-scale kinematic observations, debris) that back up the term "unstable". Similarly the term "headwall retreat" should be backed up as well. How much? What is known here? What is observed? Given the collocation

with infrastructure (cable car, ski resort) long term evidence should be available apart from regional spatial data and references.

REPLY: We decided to shift the focus away from the deglaciation aspect towards fracture kinematics and active layer dynamics. Nevertheless the comment reveals that the site description requires improvement. We will provide more information on the rockwall-glacier interaction and the site conditions in the study site description. We will no longer refer to headwall rates here, since this is beyond the scope of the data.

COMMENT: With respect to the data presented there are some issues that should be fixed: The temperature "Crack Top" is not properly described ("Crack top temperatures may not represent the entire fracture"). Where is it measured and what does this represent exactly? The air temperature and snow height that is in large parts used for process analysis is measured on the glacier, yet your rock wall is north facing above the glacier (up to 100m altitide). How do you correct this air temperature/snow height to reflect conditions inside the steep north-facing rock wall? Your precipitation measurement is from a station 500-600m lower in altitude and 2km away. This gives you an impression of the regional precipitation (sum), but in an alpine setting it is doubtful if it really captures event-by-event details w.r.t. precipitation, especially for the steep rockwall environment and strong (warm) summer liquid precipitation you are targeting. A co-located precipitation sensor would be highly beneficial here. The model developed here (section 4.3) lacks detail. In it's current state, the model cannot be reproduced.

REPLY: We will revise our measurement setup description to better illustrate our measurement. Concerning the climate data, we did not perform any corrections to cover distance and altitudinal variability. Air temperature and snow height of the climate stations are only part of the site description and not used in the analysis. We referred to precipitation events of the mentioned station 2 km away. We are not aware of any technique to correct these data, but we consider this distance relatively small compared to similar studies. For the purpose of our analysis, we focus on the timing of the events and not on the amount of rainfall. We gratefully acknowledge further comments on

ways to correct for these distance changes, but it remains to be proven how local variability of slope, sheltering or aspect, can be reproduced by a correction. Concerning the modelling approach, we will change to a different modelling approach (Weber et al. 2017) and therefore generate a new model description.

COMMENT: Last I want to comment that you regularly relate to "randkluft systems" and to people knowing your project (history) it is known that you are actively exploring/instrumenting also the perimeter of the Kitzsteinhorn rock walls below the glacier surface but this paper does not show any evidence from below the glacier surface. Therefore any connect to processes specific to this "randkluft" regime, e.g. last sentence of the abstract, middle section of section 5.3 are highly speculative at the best. Similarly, speculative statements ("other mechanisms may affect fracture dynamics" "may lead to...") do not offer any sound interpretation of what is/can be observed in this case study. The general impression that this manuscript is not quite mature yet is further exacerbated by the fact that figure captions and figures are not located together, which makes the manuscript rather hard to decipher.

REPLY: In the revised version of the manuscript we will shift the focus away from the randkluft and deglaciation. Therefore this comment will no longer apply. We will produce a more mature version of the manuscript. Dislocation of figures and captures will be avoided in a revised version. This was mainly caused by the template used.